# *Crepis vesicaria* L. subsp. *taraxacifolia* Leaves: Nutritional Profile, Phenolic Composition and Biological Properties

**DOI:** 10.3390/ijerph18010151

**Published:** 2020-12-28

**Authors:** Sónia Pedreiro, Sandrine da Ressurreição, Maria Lopes, Maria Teresa Cruz, Teresa Batista, Artur Figueirinha, Fernando Ramos

**Affiliations:** 1Faculty of Pharmacy, University of Coimbra, 3000-548 Coimbra, Portugal; soniaraquel_pedreiro@sapo.pt (S.P.); mlopes108@gmail.com (M.L.); trosete@ff.uc.pt (M.T.C.); mtpmb@ff.uc.pt (T.B.); framos@ff.uc.pt (F.R.); 2LAQV, REQUIMTE, Faculty of Pharmacy, University of Coimbra, Azinhaga de Santa Comba, 3000-548 Coimbra, Portugal; 3Polytechnic of Coimbra, Coimbra Agriculture School, Bencanta, 3045-601 Coimbra, Portugal; sandrine@esac.pt; 4Research Centre for Natural Resources, Environment and Society (CERNAS), Escola Superior Agrária de Coimbra, Bencanta, 3045-601 Coimbra, Portugal; 5CNC-Center for Neuroscience and Cell Biology, University of Coimbra, 3000-548 Coimbra, Portugal; 6CIEPQPF, FFUC, Pólo das Ciências da Saúde, Azinhaga de Santa Comba, University of Coimbra, 3000-548 Coimbra, Portugal

**Keywords:** *Crepis vesicaria* L. subsp. *taraxacifolia*, nutritional value, phenolic profile, chicoric acid, antioxidant, anti-inflammatory

## Abstract

*Crepis vesicaria* subsp. *taraxacifolia* (Cv) of Asteraceae family is used as food and in traditional medicine. However there are no studies on its nutritional value, phenolic composition and biological activities. In the present work, a nutritional analysis of Cv leaves was performed and its phenolic content and biological properties evaluated. The nutritional profile was achieved by gas chromatography (GC). A 70% ethanolic extract was prepared and characterized by HLPC-PDA-ESI/MS^n^. The quantification of chicoric acid was determined by HPLC-PDA. Subsequently, it was evaluated its antioxidant activity by DPPH, ABTS and FRAP methods. The anti-inflammatory activity and cellular viability was assessed in Raw 264.7 macrophages. On wet weight basis, carbohydrates were the most abundant macronutrients (9.99%), followed by minerals (2.74%) (mainly K, Ca and Na), protein (1.04%) and lipids (0.69%), with a low energetic contribution (175.19 KJ/100 g). The Cv extract is constituted essentially by phenolic acids as caffeic, ferulic and quinic acid derivatives being the major phenolic constituent chicoric acid (130.5 mg/g extract). The extract exhibited antioxidant activity in DPPH, ABTS and FRAP assays and inhibited the nitric oxide (NO) production induced by LPS (IC_50_ = 0.428 ± 0.007 mg/mL) without cytotoxicity at all concentrations tested. Conclusions: Given the nutritional and phenolic profile and antioxidant and anti-inflammatory properties, Cv could be a promising useful source of functional food ingredients.

## 1. Introduction

The genus *Crepis* belongs to the Asteraceae family comprising about 200 species, and is widely distributed in the Northern Hemisphere, Africa and also in South East Asia [1]. The aerial parts and roots from the plants of the genus *Crepis* are widely used in foods like salads [2], infusions [3], decoctions [4], omelettes, pasta dough and pan-fried [2]. The plants of this genus are also used in traditional medicine to treat jaundice [5], hepatic disorders [6], cardiovascular diseases [7], cough [3,6], catarrh [3], cold [3], diabetes [4], kidney stones [8], eye diseases [6], abdominal colic [6], depurative, blood cleaning, laxative and as a diuretic [2]. It can be used externally in wound healing, bruises and inflammations [8].

Biological properties and phenolic composition have been evaluated for some species. Aerial part and root extracts of *Crepis foetida* L. subsp. *rhoeadifolia* showed antioxidant activity in DPPH and thiobarbituric acid reactive substances (TBARS) assays. In these extracts phenolic compounds to which antioxidant activity has been attributed were identified, namely chlorogenic acid and luteolin in the aerial parts and chlorogenic acid in the roots [7]. A methanolic extract from the flowers of this species showed that chlorogenic acid was the major phenolic compound. This extract presented high antiproliferative activity in HEPG-2, Caco-2, MCF-7 and MCF-10A cells, antioxidant (in DPPH, ABTS, nitric oxide and superoxide radical scavenging assays), anticholinesterase and antityrosinase activities [9]. Aqueous and ethanolic extracts of *Crepis japonica* L. showed antiproliferative activity against leukemia and sarcoma. Moreover, the ethanolic extract presented antiviral activity against respiratory syncytial vírus (anti-RSV). The phenolic content, including hydrolysable tannins may be responsible for this activity [10]. The 3,4-dicaffeoylquinic acid, 3,5-dicaffeoylquinic acid and luteolin-7-*O*-glucoside were isolated from the *Crepis japonica* ethanolic extract. The first two compounds exhibited significant anti-RSV activity. Moreover these three compounds together showed some antibacterial activity against *Vibrio cholerae* and *Vibrio parahaemolyticus* [11]. Besides these activities in this plant, anti-inflammatory, immunosuppressive, antiallergic, antioxidant, analgesic, central nervous system depressant and nematicidal activities were also reported [12].

It is known that reactive oxygen species (ROS), when unregulated, are related to several pathologies such as inflammation, through NF-kB signaling pathway activation [13]. The NF-kB transcription factor is present in the cytoplasm and is in its inactive state due to its association with the inhibitor complex of nuclear factor kappa B kinase (IKK). When this kinase is activated, NF-kB is released and enters the nucleus by activating the transcription of a variety of genes that participate in inflammatory and immune responses [14], such as interleukines IL-1β, IL-6 and IL-8, tumor necrosis factor-α (TNF-α) [15], prostaglandins [16], chemokine ligand 5 (CCL5), transcription of inducible nitric oxide synthase (iNOS), leading to the production of nitric oxide (NO) [17], and cyclooxygenase-2 (COX-2) [18], among others. Phenolic compounds such as phenolic acids, flavonoids and tannins have been identified as good radical scavengers [17]. The mechanisms by which they act in the radical scavenging are involved in signaling pathways of inflammation activation [13]. Phenolic acids presented anti-inflammatory activity acting mainly at the level of the proteasome, inhibiting this and, also, the activation of the NF-kB, since it maintains the phosphorylation levels of IkBα. These mechanisms were attributed to the action of chlorogenic acid, since this was identified as main phenolic acid together with *p*-coumaric acid derivatives [17]. These derivatives have been found to have the ability to inhibit iNOS-dependent NF-kB and COX-2 expression [19]. Phenolic acids from *Lippia* genus inhibited the carrageenan-induced pro-inflammatory cytokine production, namely IL-1β, IL-33 and TNF-α and consequently suppressed the NF-kB activation [16]. Polyphenols from *Ilex latifolia* Thunb. ethanolic extract showed high antioxidant and anti-inflammatory activities, through decreasing the production of NO, COX-2 and pro-inflammatory cytokines via inhibitions of MAPKs, namely ERK and JNK, and NF-kB activation [20]. In LPS-induced acute lung injury rats model, chlorogenic acid decreased the activity of iNOS and suppressing the activation of NF-kB [16]. Others phenolic compounds decreased TLR-4 upregulation, NOX activation and NF-kB activation in LPS-induced renal inflammation rat model [16]. The antioxidant and anti-inflammatory activities of rice bran (RB) phenolic compounds were evaluated in human umbilical vein endothelial cells (HUVECs) with induced oxidative stress. The RB extract regulated antioxidant genes, namely Nrf2, NQO1, HO1 and NOX4, as well anti-inflammatory genes (ICAM1, eNOS, CD39 and CD73). This activities were attributed to synergistic interactions between phenolic acids including *p*-coumaric acid, vanillic acid, caffeic acid, ferulic and syringic acid [21]. These studies support the correlation between oxidative stress and inflammation as well the biological effects of phenolic compounds on these.

There are no studies on phytochemical composition and biological activities of *Crepis vesicaria* L. subsp. *taraxacifolia* (Cv). Commonly known as beaked hawk’s beard [2], this plant is used traditionally in foods and the treatment of diverse ailments. The cooking water of Cv young leaves are traditionally used as depurative, blood cleaning, diuretic and laxative [2]. Therefore it’s important to study this phytoconstituents and to evaluate its health impact. In the present work, there was evaluated the phenolic and nutritional composition of Cv, as well assessed the antioxidant and anti-inflammatory activities.

## 2. Material and Methods

### 2.1. Plant Material and Extract Preparation

*Crepis vesicaria* L. subsp. *taraxacifolia* (Cv) plant was collected and identified by J. Paiva (Botany Department, University of Coimbra, Coimbra, Portugal). A voucher specimen (A. Figueirinha, 175) was deposited in the herbarium of the University of Coimbra, Faculty of Pharmacy. The leaves, dried in a circulating air drying oven, were milled in a knife mill (KSM 2, BRAUN, Frankfurt, Germany), avoiding the overheating of the sample, and sieved through a 60 mesh sieve. Subsequently, extracts were prepared from the powdered material with different solvents in a proportion of 1:100 (*w*/*v*). In order to improve the extraction of more active compounds, several extractions of Cv leaves (10 mg of dry plant/mL) were made with ethanol/water in various grades: 10%, 30%, 55%, 70% and 100% EtOH. The results of three independent assays showed that the reduction percentage of DPPH radical for the different extracts at 0.33 mg/mL was: 100% EtOH (28.22 ± 0.1575%) < 15% EtOH (66.92 ± 1.083%) < 30% EtOH (91.83 ± 0.602%) < 55% EtOH (90.96 ± 2.205%) < 70% EtOH extract (91.87 ± 2.066%). Thus, a leaves infusion of Cv (ICv) prepared according to its ethomedicinal uses, and 70% ethanol extracts were obtained. These extracts were filtered under vacuum, concentrated in a rotavapor at 40 °C, frozen, freeze-dried and kept at −20 °C in the dark until use. In the leaves 70% ethanol Cv extract (Cv 70% EtOH), a yield of 22.85% of dry plant was obtained. Relatively to ICv the yield of dry plant obtained was 30.1%. Both extracts were rich in soluble phenolics.

### 2.2. Chemical Characterization

#### 2.2.1. Nutrient Composition Analysis

Proximate composition parameters were measured according to the international standards methods of Official Methods of Analysis of AOAC International [22], except neutral detergent fiber (NDF), acid detergent fiber (ADF) and acid detergent lignin (ADL) for calculation of cellulose, hemicellulose and lenhin, respectively [23]. Moisture evaluation was performed by oven drying sample at 105 °C until constant weight. Protein was determined by Kjeldahl method, using a protein conversion factor of 6.25. Lipids were gravimetrically quantified after a continuous extraction process in a Soxhlet apparatus by diethyl ether. Fatty acids were analysed by gas phase chromatography (GC-FID) of fatty acid methyl esters, and the quantification was performed by Supelco standards (Sigma-Aldrich, St Louis, MO, USA). The total dietary fiber, soluble and insoluble dietary fiber contents were determined using the Supelco enzyme kit TDF100A (Sigma-Aldrich). Crude fiber was gravimetrically quantified after chemical digestion and solubilisation of other materials. The fiber residue weight was then corrected for ash content. Ash was obtained by incineration of all organic matter of the sample in a muffle furnace at 550 °C. The Nitrogen-free extractives were estimated, considering the following equation: Nitrogen-free extractives = 100 − (moisture + ash + lipids + protein + crude fiber). The total carbohydrates were estimated, considering the following equation: Total carbohydrates = 100 − (moisture + ash + lipids + protein). Quantification and sugars were performed by High Performance Liquid Chromatography with refractive index detection (HPLC-RI). The separation column used was HC-75 Ca^2+^ 305 × 7.8 mm (Hamilton, Energy Way Reno, NV, USA) with ultrapure water with traces of sodium azide mobile phase, at a flow rate of 0.6 mL/min, at 80 °C. The quantification was performed by BioUltra standards (Sigma-Aldrich). The available carbohydrates were estimated, based on the following equation: Available Carbohydrates = 100 − (moisture + ash + lipids + protein + dietary fiber). Quantification of sugars were performed by high performance liquid chromatography (HPLC), using BioUltra standards (Sigma-Aldrich). Energy values are expressed in Kcal and KJ/100 g and were calculated according to Regulation (EU) n° 1169/2011 of the European Parliament and of the Council of 25 October 2011. Minerals were determined by flame atomic absorption spectrometry (FAAS), with the exception of cadmium and lead traces, which were determined by graphite furnace atomic absorption spectrometry (GFAAS). Mercury traces was analysed by an AMA254 Mercury Analyzer (Leco, St Joseph, MI, USA) and phosphorus, by spectrophotometry.

#### 2.2.2. Phenolic Profile HPLC-PDA-ESI/MS^n^

The phenolic profile of Cv (ethanol 70% extract) was carried out on a liquid chromatograph (Finnigan Surveyor, THERMO, Waltham, MA, USA) with a Spherisorb ODS-2 column (150 × 2.1 mm i.d.; particle size, 3 µm; Waters Corp., Milford, MA, USA) and a Spherisorb ODS-2 guard cartridge (10 × 4.6 mm i.d.; particle size, 5 µm; Waters Corp. Milford, MA, USA). The separation occurred at 25 °C with a mobile phase constituted by 2% aqueous formic acid (v/v) (A) and methanol (B) in a discontinuous gradient of 5–15% B (0–10 min), 15–25% B (10–15 min), 25–50% B (15–40 min), 50–80% B (40–50 min), followed by an isocratic elution (50–60 min), a gradient 80–100% B (60–65 min) and other isocratic elution for 5 min, at a flow rate of 200 µL/min.

The first detection was done with a PDA detector ((Finnigan Surveyor, THERMO, Waltham, MA, USA)) at a wavelength range 200–400 nm, followed by a second detection using an Linear Ion Trap Mass Spectrometer (LIT-MS) (LTQ XL, Thermo Waltham, MA, USA). Mass spectra were obtained in the negative ion mode. The mass spectrometer acquired three consecutive spectra: full mass (*m*/*z* 125–1500), MS^2^ of the most abundant ion in the full mass and MS^3^ of the most abundant ion in the MS^2^. Source voltage was 4.5 kV and the capillary temperature and voltage were 250 °C and −10 V, respectively. The sheath and auxiliary gas used was nitrogen at 20 Finnigan arbitrary units with helium as collision gas with a normalized energy of 45%. XCALIBUR software (Thermo, Waltham, MA, USA) was used for data treatment.

#### 2.2.3. Quantification by HPLC-PDA

Quantification of L-chicoric acid in Cv 70% EtOH was performed in a chromatograph with a photodiode array (PDA) detector (Gilson Electronics SA, Villiers le Bel, France). The analysis were performed on a Spherisorb S5 ODS-2 column (250 × 4.6 mm i.d., 5 µm) (Waters Milford, MA, USA) with a C18 guard cartridge (30 × 4 mm i.d., 5 µm) (Nucleosil, Macherey-Nagel, Düren, Germany), at 24 °C. The mobile phase was constituted by methanol 100% (B) and formic acid 5% (A). The elution was made at a flow rate of 1 mL/min. The gradient used was: 5–15% B (0–10 min), 15–25% B (10–15 min), 25–50% B (15–40 min), 50–80% B (40–50 min) followed by an isocratic elution of 80% B (50–60 min), 80–100% B (60–70 min) and finally, an isocratic elution of 100% B (70–85 min). The volume of the sample injected was 100 µL. The UV-Visible spectra acquisition was performed between 200–600 nm and the chromatographic profiles were recorded at the wavelengths 280 and 320 nm. Data treatment was carried out with Unipoint^®^, version 2.10 software (Gilson, Middleton, WI, USA).

Quantification of the L-chicoric acid was performed using commercial standard dissolved in methanol (10 to 150 µg/mL) as external standard L-chicoric acid (Sigma Aldrich St. Louis, MO, USA). The chicoric acid present in the Cv 70% EtOH extract was quantified by the absorbance recorded in the chromatogram relative to this standard (330 nm). Three independent injections (100 µL) were performed in duplicate for each sample. The least-squares regression model was used to assess the correlation between peak area and concentration. The detection (LOD) and quantification (LOQ) limits were calculated from the calibration curve. The quantification of the chicoric acid in Cv 70% EtOH extract (identified first by HPLC-PDA-MS^n^) was made using the standard calibration curve and the peak area of the compound.

### 2.3. Antioxidant Activity

#### 2.3.1. 2,2-Diphenyl-1-Picrylhydrazyl Radical Assay (DPPH)

Free radical-scavenging activity of the infusion and ethanol/water Cv extracts was evaluated using the *DPPH* method previously described [24]. Briefly, aliquots of samples (100 µL) were assessed by their reactivity with methanolic solution of 500 µM *DPPH* (500 µL) in the presence of 100 mM acetate buffer, pH 6.0 (100 µL). The reaction mixtures (300 µL) were kept for 30 min at room temperature, in the dark. The decreases in the absorbance were measured at 517 nm, in a Thermo scientific multiskan FC plate reader. The % of reduction of *DPPH* of the Cv extracts were determined by:(1)$$\%\;reduction\;of\;DPPH=100-\frac{Abs\;sample-Abs\;control}{Abs\;control}$$

Posteriorly, the obtained values were plotted in a graph % of DPPH reduction vs. concentration in µg/mL. The IC_50_ was interpolated in the graph for the correspondent value of 50% reduction.

The results were expressed as Trolox equivalent (TE), defined as the concentration of the extract with antioxidant capacity equivalent to 1 mM of Trolox solution. This value was obtained interpolating the absorbance of 1 mM Trolox in the graph % of DPPH reduction vs. concentration. All the determinations were performed in triplicate.

#### 2.3.2. Ferric Reducing Antioxidant Power Assay (FRAP)

Ferric reducing ability was evaluated with slight modifications [25]. The FRAP reagent was prepared by mixing 300 mM acetate buffer, 10 mL TPTZ (Sigma–Aldrich St. Louis, MO, USA) in 40 mM HCl and 20 mM FeCl_3_.6H_2_O (Merck, Darmstadt, Germany) in the proportion of 10:1:1 (*v*/*v*/*v*). The extract (100 µL) was added to 3 mL of the FRAP reagent. An intense blue color complex was formed when ferric tripyridyl triazine (Fe^3+^ TPTZ) complex was reduced to ferrous (Fe^2+^) form. The absorbance was measured at 593 nm, against a reagent blank, after incubation at room temperature for 6 min. The results were expressed as trolox equivalent (TE) values obtained using a calibration curve for Trolox (31.25–1000 mM). All the determinations were performed in triplicate.

#### 2.3.3. 2,2′-Azinobis-(3-ethylbenzothiazoline-6-sulfonate) Assay (pH = 7) (ABTS)

The ABTS assay described by [26], consisted in generating the ABTS•+ radical by the oxidation of ABTS (7 mM) with potassium persulphate (2.45 mM) (Merck) in water. After 12–16 h in dark and at room temperature, this solution was diluted with phosphate buffered saline (PBS) at pH 7 to give an absorbance of 0.7 ± 0.02 at 734 nm. The extract (50 µL) was mixed with 2 mL of the ABTS + solution and vortexed for 10 s. After 4 min of reaction, the absorbance was measured at 734 nm. The IC_50_ value was interpolated in a graph % of ABTS reduction vs. concentration in µg/mL for the correspondent value of 50% reduction. The results were expressed as TE, obtained interpolating the absorbance of 1 mM trolox in the graph % of ABTS reduction vs. concentration. Three independent experiments in triplicate were performed for each of the assayed extracts.

### 2.4. Anti-Inflammatory Activity Evaluation

#### 2.4.1. Nitrite Production by Griess Assay

Raw 264.7, a mouse leukemic monocyte macrophage cell line from American Type Culture Collection (Manassas, VA, USA), and kindly supplied by Dr. Otília Vieira (Center for Neurosciences and Cell Biology, University of Coimbra, Portugal), was cultured in Iscove’s Modified Dulbecco’s Eagle medium supplemented with 10% non-inactivated fetal bovine serum, 100 U/mL penicillin, and 100 µg/mL streptomycin at 37 °C in a humidified atmosphere of 95% air and 5% CO_2_. The cells were monitored to detect any morphological change. For the experiments, the cells were plated (0.6 × 10^5^ cells/well) with culture medium and allowed to stabilize for 12 h. Then the cells were incubated during 24 h at 37 °C in culture medium (control) or stimulated with 1 µg/mL of bacteria lipopolysaccharide (LPS) (Sigma) with or without different concentrations of extract (0.1–2.0 mg/mL).

The anti-inflammatory activity was determined by the nitric oxide production, measured indirectly by the accumulation of nitrite in the supernatant through a colorimetric assay with Griess reagent [0.1% (*m*/*v*) of N-(1-naphthyl)-ethylenediamine dihydrocloride and 1% (*m*/*v*) of sulfanilamide with 5% of phosphoric acid] [27]. To perform the assay, it was used 100 µL of the supernatant and 100 µL of Griess’s reagent and then stored away from light during 30 min. The absorbance at 550 nm was measured in an automated plate reader (Synergy HT, BioTek Instruments SAS, Colmar, France). Culture medium was used as blank and nitrite concentration was determined from a regression analysis using serial dilutions of sodium nitrite as standard.

#### 2.4.2. Assessment of Cell Viability by Resazurin Assay

In order to evaluate the cytotoxicity it was performed the resazurin assay [28]. After the incubation with the samples, the cells were incubated with 100 µL of a resazurin solution (10 µM in culture medium) during 2 h at 37 °C in a humid atmosphere with 5% CO_2_/95% air. Quantification of resorufin was performed using a plate reader (Synergy HT, BioTek, Instruments SAS, Colmar, France) at 570 nm, with an optical filter for 620 nm.

### 2.5. Statistical Analysis

All samples were analysed, at least, in triplicates and the results were expressed as mean ± standard deviation (SD). To calculate the IC_50_ values for the anti-inflammatory activity, the linearization of the dose-response curve was performed as described by Chou [29].

The statistical analysis of the cellular viability and anti-inflammatory activity was performed in GraphPad Prism program (version 5.02, GraphPad Software, San Diego, CA, USA). For the comparison between treatment conditions and control it was used two-sided unpaired *t*-test. To evaluate the effect of different treatments to LPS-stimulated cells it was performed One-way ANOVA followed by Bonferroni’s test. The limit of significance was set at *** *p* < 0.001.

## 3. Results and Discussion

### 3.1. Nutrient Composition of C. vesicaria

The knowledge of the nutritional properties of wild plants is crucial to assess their suitability for human consumption. In this study, the nutritional profile of *Crepis vesicaria* subsp. *taraxacifolia* leaves was analyzed.

#### 3.1.1. Nutritional Analysis of *Crepis vesicaria* subsp. *taraxacifolia* Leaves

The nutritive content of Cv leaves was determined (Table 1). The proximate composition revealed high moisture content, even though all foods contain water; those with a higher content are more prone to the rapid occurrence of microbial spoilage phenomena, enzymatic degradation and other moisture-dependent chemical deterioration reactions. Therefore, precautions should be considered to prevent rapid deterioration during storage, such as drying or freezing.

Total carbohydrates, calculated by difference, were the most abundant macronutrients (9.99 g/100 g wet weight (*w*/*w*)), followed by ash, protein and lipids. Carbohydrates play a major role in human diet. They are the main source of energy, and also help to maintain glycemic homeostasis and gastrointestinal integrity, among other functions. A healthy adult diet should include about 130 g of carbohydrates per day [30]. Cv leaves contain an important amount of carbohydrates, which is in line with what has been reported for other wild Asteraceae plants traditionally consumed in the Mediterranean region, such as *Taraxacum obovatum*, *Chondrilla juncea*, *Sonchus oleraceus*, *Cichorium intybus*, *Scolymus hispanicus* and *Silybum marianum* [31]. An important fraction of the total carbohydrates content in Cv leaves is fiber. In this study, different fiber measurement methods were used and the results showed that the chosen method has an impact on the values observed for different fiber parameters. Weende’s crude fiber analysis determines cellulose, lignin and some hemicellulose, pectin, gums and mucilages. The acid detergent lignin (ADL) measures lignin, the acid detergent fiber (ADF) determines cellulose and lignin, and the neutral detergent fiber (NDF) consists mainly in the measurement of cellulose, hemicelluloses and lignin [32]. Regardless of the method, the results reveal that Cv leaves are an interesting dietary fiber source, with insoluble dietary fiber being the major fraction. It is well established that the daily consumption of about 25–30 g of fiber, for an adult, markedly reduces the risk of cardiovascular and digestive diseases [30]. Also, Cv leaves may contain insoluble-bound phenolics present in the cell wall plant components. These insoluble-bound form can contribute for to protection of cardiovascular health [33]. Thus, the use of this plant, either individually or added to other foods, may contribute to a desired increase in fiber intake with the associated health benefits.

With regard to the available carbohydrates, the estimated value was 5.75 g/100 g (*w*/*w*). The total sugars content found was 3.76 g/100 g (*w*/*w*), with maltose as the main sugar (2.47 g/100 g, *w*/*w*), followed by fructose and glucose. Protein makes up 1.04 g/100 g, *w*/*w* of Cv leaves. This value is considerably lower than that reported by Barnett and Crawford [34]. Variations in protein levels may be due to differences between species, environmental and climatic factors, or a mixture of both.

#### 3.1.2. Lipid and Fatty Acids Composition of *Crepis vesicaria* subsp. *taraxacifolia* Leaves

According to Table 2, the lipid content was moderate, 0.69 g/100 g, *w*/*w* (4.78 g/100 g dry weight (dw)), higher than that reported for *C. Juncea*, the highest lipid content Asteraceae (0.79 g/100 g, *w*/*w*) analyzed in the study of García-Herrera [31]. The fatty acid profile of Cv leaves showed a predominance of polyunsaturated fatty acids (PUFA) (402.84 mg/100 g, *w*/*w*), mainly comprised by α-linolenic acid (343.24 mg/100 g). Total saturated fatty acids (SFA) concentration was 159.82 mg/100 g, *w*/*w*, with the main constituent being palmitic acid (108.75 mg/100 g, *w*/*w*). For a nutritional “good quality”, including beneficial effects in terms of cardiovascular risk reduction, the PUFA/SFA ratio should be > 0.45, whilst n-3/n-6 fatty acids ratio should be > 4 [35]. In the present study, the PUFA/SFA ratio was 2.52 and the n-3/n-6 fatty acids ratio was 5.76. The presence of considerable amounts of oleic acid (60.49 mg/100 g, *w*/*w*) should also be highlighted, given the beneficial properties that have been attributed to it in the context of the immunomodulation, prevention and treatment of several pathologies such as cancer, cardiovascular and autoimmune diseases, and metabolic disturbances [36].

#### 3.1.3. Minerals and Heavy Metal Composition of Cv Leaves

Given the results in Table 3, *Crepis vesicaria* L. subsp. *taraxacifolia* leaves exhibited moderate levels of ash (2.74 g/100 g, *w*/*w*). This value is within the recommended range for human consumption and reveals considerable mineral richness, which is corroborated by studies on similar species, such as *C. commutata* and *C. vesicaria* [37]. The mineral fraction is an aspect of greater relevance in the use of edible plants in human nutrition. The inappropriate intake of minerals (macrominerals and trace minerals) is the cause of multiple degenerative and chronic diseases. Calcium (Ca), phosphorous (P), magnesium (Mg), sodium (Na), potassium (K) and iron (Fe) are essential elements and their intake is necessary at mg/kg level to keep the human body healthy. Zinc (Zn), copper (Cu), manganese (Mn), chromium (Cr), and nickel (Ni) are required at trace levels in the diet [38]. Concerning the macrominerals composition of Cv leaves, K, Ca and Na were the most abundant (591.29 mg/100 g, *w*/*w*; 309.93 mg/100 g, *w*/*w*; 76.78 mg/100 g, *w*/*w*, respectively). The macromineral profile found was identical to that reported for *C. vesicaria* (K > Ca > Na > P > Mg), but different from that observed in *C. commutata* (K > P > Na > Mg > Ca) [37]. When compared to other wild Asteraceae, such as *S. hispanicus*, K and Ca contents are lower, but Cv can still be considered a remarkable source of these minerals, better than many conventional vegetables [31]. Zn and Mn were found as major trace minerals (0.86 mg/100 g, *w*/*w* and 0.83 mg/100 g, *w*/*w*, respectively). The most abundant trace minerals in *C. vesicaria* and *C. commutate* were Fe and Mn. *Crepis* spp. seem to be a good source of Mn. The contaminants cadmium (Cd), lead (Pb) and mercury (Hg) were detected. These toxic metallic elements can induce damage to multiple organs and have carcinogenic effects [39]. Pb levels are below the maximum values legislated, 0.30 mg/kg, *w*/*w*. However, the concentration of Cd coincides with the maximum level of contamination that is considered safe, 0.2 mg/kg, *w*/*w* [40]. Overall, the results indicate that, when located in polluted areas, these plants can accumulate toxic metals in concentrations that may represent a risk to the consumer’s health.

### 3.2. Screening for Antioxidant/Scavenging Activity

The ability of ROS to activate the inflammation signaling pathway, through activation of pro-inflammatory cytokines is well known. The literature describes that colorimetric methods to assess antioxidant activity like DPPH and ABTS, are a good tool to select the extracts more promising [41]. Also, it was reported that phenolic extracts bearing higher radical scavenging towards DPPH and ABTS, present higher inhibition of NF-kB activation mediated by ROS [41]. Given the correlation between antiradical activity and inhibition of the NF-kB signaling pathway, the antioxidant activity of the extracts was screened using the DPPH and ABTS colorimetric methods. Based on the results obtained, it was chosen the extract that demonstrated the greatest activity in these tests.

The infusion (10 mg of dry plant/mL) was screened for antioxidant activity as it is the form of use in traditional medicine. However, the percentage of reduction observed was 20.27%. As the Cv 70% EtOH extract was the most active extract it was lyophilized (previously described in Material and Methods) and characterized relatively to its phenolic profile, and antioxidant and anti-inflammatory activities.

Regarding antioxidant activity, the infusion presented an IC_50_ of 103.22 ± 5.61 µg/mL and a TEAC of 441.980 ± 0.058 mg/mL. The IC_50_ of Cv 70% EtOH was 26.20 ± 1.86 µg/mL and TEAC of 111.980 ± 0.041 mg/mL meaning that this extract is more active than the infusion. Subsequently, the antioxidant activity of the 70% ethanol extract was assessed by FRAP and ABTS methods (Table 4). The ABTS and DPPH methods are based on electron and H atom transfer while the FRAP is based on electron transfer reaction [42]. Attending to the results, the Cv extract present reducing power besides their ability in scavenging free radicals. The results shown that the Cv 70% EtOH extract has a good radical-scavenging activity and antioxidant activity.

### 3.3. Phenolic Profile of 70% Ethanolic Extract from Crepis vesicaria subsp. taraxacifolia

Based on the given results relatively to the antioxidant activity, the 70% EtOH from Cv extract is the most active. Therefore, the phenolic profile by HPLC-PDA-MS^n^ of this extract was assessed (Figure 1).

According to the absorption spectra, the 70% EtOH Cv extract is mainly composed of phenolic acids, generally presenting a shoulder at 295 nm and a maximum wavelength of 330 nm (Table 5), indicating to be caffeic or ferulic acids derivatives [43].

In an attempt to identify the compounds of this extract, HPLC-PDA-ESI/MS^n^ was performed. The results (Table 5) showed that the extract consisted mainly of phenolic acids, namely caffeic and ferulic acid derivatives as well as chicoric acid isomers. The chicoric acid was identified as the main compound of the Cv 70% ethanol extract.

Compound **1**. MS analysis showed a molecular ion [M-H]^-^ at *m*/*z* 179 and a fragmentation pattern typical of caffeic acid. MS^2^ data presented fragments at *m*/*z* 135 indicating a decarboxylated caffeic acid moiety [(M-H-CO_2_]^-^. The compound 1 was tentatively identified as caffeic acid [44].

Compound **2**. This compound presents a molecular ion [M-H]^-^ at *m*/*z* 191. MS^2^ most abundant fragments are *m*/*z* 173 indicating a dehydrated quinic acid moiety [M-H-H_2_O]^-^. This compound was tentatively identified as quinic acid [44].

Compounds **3**, **4** and **5**. The molecular ion [M-H]^-^ occurs at *m*/*z* 473. The MS^2^ presents a fragment at *m*/*z* 311, indicating the presence of deprotonated caftaric acid [M-H-C_13_H_12_O_9_]^-^ and *m*/*z* 293 corresponding to the neutral loss of caffeic acid [M-H-C_9_H_8_O_4_]^-^. MS^3^ profiles have a fragment at *m*/*z* 149 corresponding to the tartaric acid and at *m*/*z* 179 corresponding to a deprotonated molecule of caffeic acid [M-H-C_9_H_8_O_4_]^-^. Based on this fragmentation pattern, these compounds were tentatively identified as chicoric acid isomers [47]. Accordingly with the literature, the most abundant chicoric acid isomer is L-chicoric acid [47]. The quantification by HPLC-PDA of this isomer was performed using a standard. The peak of the L-chicoric acid standard has approximately the same retention time of peak 5. Therefore, peak 5 was tentatively identified as L-chicoric acid.

Compound **6**. This compound has a molecular ion [M-H]^-^ at *m*/*z* 487. MS^2^ showed fragments at *m*/*z* 325 (loss of 162 Da) indicating the loss of a hexose [M-H-C_6_H_12_O_6_]^-^ and at 307 that indicates the subsequent loss of water [M-H-C_6_H_12_O_6_-H_2_O]^-^. The MS^3^ fragment *m*/*z* 193 indicates the presence of ferulic acid [48], that can be probably associated to a hexosylpentosyl residue. All the data suggest that compound **6** was tentatively identified as feruloyl hexosylpentoside [46].

### 3.4. Quantification of Chicoric Acid

The major constituents in *Crepis vesicaria* subsp. *taraxacifolia* were chicoric acid derivatives and the evaluated activities were attributed to these compounds. Therefore, the L- chicoric acid was quantified by HPLC-PDA. The equation of calibration curve of L-chicoric acid was y = 4011236.2307 × −36474316.1324 (r^2^ = 0.99). Based on this equation, the concentration of L-chicoric acid in the Cv 70% EtOH extract was 130.5 ± 4.2 mg/g extract of Cv 70% EtOH. The LOD and LOQ were 19.74 ± 3.33 mg/g extract and 44.58 ± 2.96 mg/g extract, respectively.

### 3.5. Assessment of Cell Viability of the Cv 70% EtOH Extract

The citotoxicity of the Cv 70% EtOH extract in macrophages was evaluated. The results (Figure 2) showed that none of the tested concentrations were cytotoxic.

The cytotoxicity of chicoric acid in macrophages (Raw 264.7 cells) has also been was tested. The studies performed have shown that this compound is not cytotoxic [49]. There are few cell viability studies in normal cells with Cv extracts. However, some researchers studied the effects of a methanol extract of Cv flowers on tumor (HEPG-2, Caco-2 and MCF-7) and non-tumor (MCF-10A) cells. The Cv extract was not cytotoxic to the non-tumor cell line and cytotoxic to the tumor lines and therefore had some selectivity over tumor cells [9].

### 3.6. Antioxidant and Anti-Inflammatory Activity of the Cv 70% Ethanol Extract

The results showed that Cv 70% ethanol extract inhibited NO production in a dose-dependent manner (Figure 3) and the IC_50_ was 0.428 ± 0.00669 mg/mL. Cv extract is little active in inflammation compared to the results obtained in antioxidant activity.

Reactive oxygen species are involved in various pathologies including inflammation. Cv extract was shown to have a high antioxidant activity. According to the characterization of the extract by HPLC-PDA-ESI/MS^n^, the extract has phenolic acids namely caffeic and ferulic acid derivatives as well as chicoric acid isomers. The mechanisms involved in the antioxidant activity by which phenolic compounds act are based on: ability to chelate metals, such as copper and iron, that participate in the Fenton reaction generating hydroxyl radicals; interrupt signaling pathways triggered by free radicals; interfere with enzyme activity [50]. It is known that the antioxidant activity of phenolic compounds is directly related to the number of hydroxyl groups. Chicoric acid was identified as the major compound present in the extract. This compound has two caffeic acid moieties which are responsible for the high activity observed relatively to caffeic acid [51]. Some authors relate high molecular weight phenolic compounds, such as chicoric acid, with antioxidant activity [52]. Therefore, the observed antioxidant activity by H transfer and electron transfer reaction can be mostly attributed to chicoric acid.

Other plants of the genus *Crepis* are reported to have various biological activities including anti-inflammatory activity [6]. Chicoric acid has been reported to have that activity inhibiting activated immune cells, nitric oxide synthase and cyclooxygenase-2 through its inhibitory effects on nuclear factor NF-κB and TNF-α [53,54,55]. However, the results of Cv 70% EtOH extract weren’t satisfactory in the anti-inflammatory activity. This fact can be due to the extract matrix [53] or antagonistic interactions between the matrix compound [56]. Moreover, this extract have chicoric acid isomers, and the observed activity can be due to this isomers. Further studies are needed to understand the inherent mechanisms of these compounds in anti-inflammatory activity.

## 4. Conclusions

The 70% ethanol *Crepis vesicaria* subsp. *taraxacifolia* leaves extract presents antioxidant and anti-inflammatory activities. Besides its biological activities, the Cv leaves extract demonstrated to have a high content of lipids and fatty acids. Given the observed antioxidant activity, Cv extract may be used as a functional food to prevent oxidative stress and its associated pathologies. In fact, Cv leaves displayed an interesting nutritional composition, with a low energetic contribution of 41.84 kcal/100 g, *w*/*w*. Some potential uses for this plant’s leaves may be the development of new additives for human and/or animal consumption or food supplements that contribute to a balanced diet. In a context of food insecurity, their incorporation as an ingredient in recipes may also be of interest to increase their nutritional and functional value. From another perspective, to increase the consumption of this abundant and under-exploited plant, it is important to investigate its nutritional value and antioxidant and anti-inflammatory properties, but also to ensure that its consumption is safe, i.e., without neglecting the risk of contamination by toxic metals.

## Figures and Tables

**Figure 1 ijerph-18-00151-f001:**
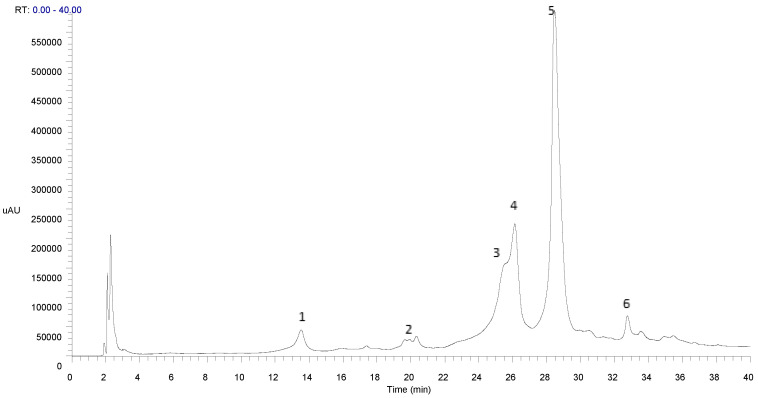
HPLC-PDA-ESI/MS^n^ profile of 70% ethanol extract from Cv recorded at 280 mn. It was used the gradient 2 described in Material and Methods section. (The chromatogram of the extract is not shown up to 40 min as no further compounds were eluted after this time period. Peaks 1–6 identification is showed in Table 5).

**Figure 2 ijerph-18-00151-f002:**
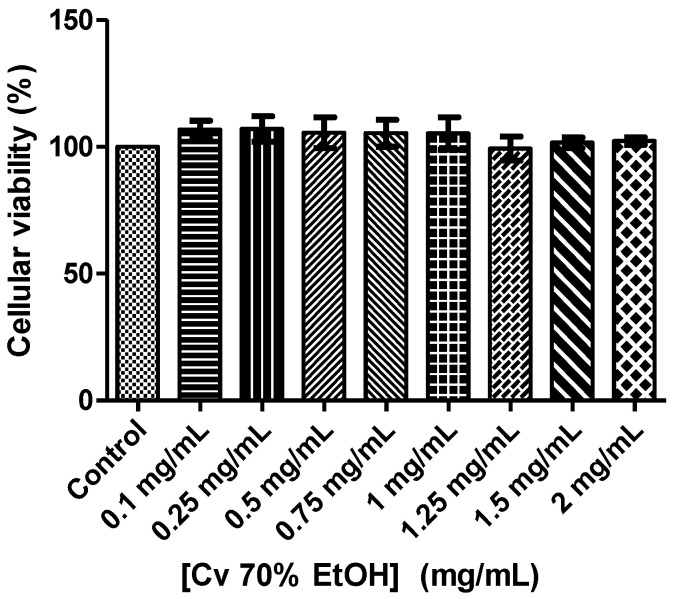
Effect of *Crepis vesicaria* subsp. *taraxacifolia* ethanolic extract on macrophages cell viability (RAW 264.7 cells). Each result represents the mean ± SD (minimum of three independent assays, performed in duplicate). The statistical tests were performed with *p* < 0.05 compared to control.

**Figure 3 ijerph-18-00151-f003:**
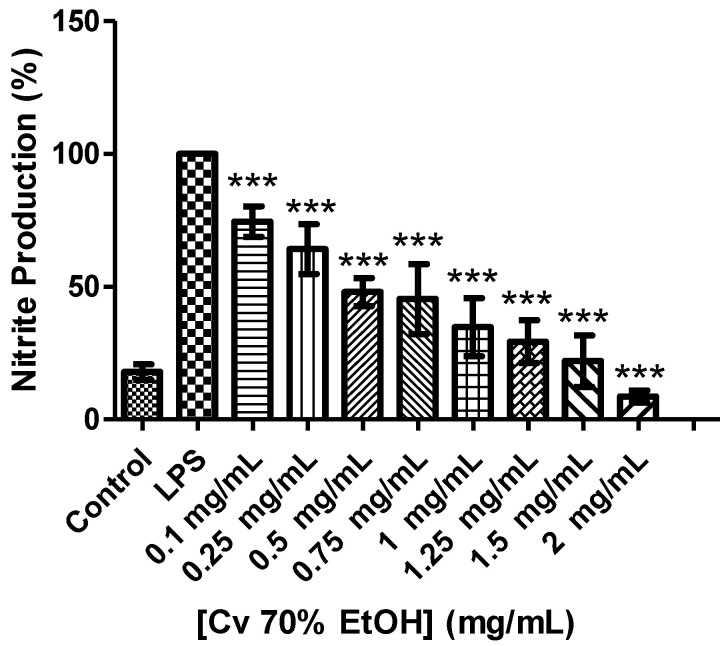
Effect of *Crepis vesicaria* subsp. *taraxacifolia* ethanolic extract on NO production in macrophages (RAW 264.7 cells). Each result represents the mean ± SD (minimum of three independent assays). *** *p* < 0.001 compared with the LPS group.

**Table 1 ijerph-18-00151-t001:** Nutritive content of *Crepis vesicaria* subsp. *taraxacifolia* leaves (mean ± SD; *n* = 3).

Composition		Raw Matter	Dry Matter
Energy (KJ/100 g)		175.190 ± 0.259	1211.80 ± 2.11
Energy (Kcal/100 g)		41.840 ± 0.062	289.43 ± 0.50
Moisture (g/100 g)		85.540 ± 0.006	-
Protein (g/100 g)		1.040 ± 0.003	7.18 ± 0.02
Dietary fiber (g/100 g)		4.240 ± 0.015	29.35 ± 0.11
Insoluble Dietary fiber (g/100 g)		3.490 ± 0.026	24.14 ± 0.18
Acid detergent fiber (ADF) (g/100 g)		3.120 ± 0.011	21.59 ± 0.08
Cellulose (g/100 g)		2.550 ± 0.002	17.61 ± 0.02
Crude fiber (g/100 g)		2.460 ± 0.009	17.00 ± 0.06
Hemicellulose (g/100 g)		0.620 ± 0.012	4.27 ± 0.09
Lignin (g/100 g)		0.440 ± 0.013	3.03 ± 0.09
Acid detergent lignin (ADL) (g/100 g)		0.430 ± 0.006	2.99 ± 0.04
Nitrogen-free extractives (g/100 g)		7.530 ± 0.010	52.11 ± 0.07
Carbohydrates	Maltose (g/100 g)	2.470 ± 0.015	17.11 ± 0.11
	Fructose (g/100 g)	0.940 ± 0.012	6.53 ± 0.08
	Glucose (g/100 g)	0.340 ± 0.012	2.37 ± 0.08

**Table 2 ijerph-18-00151-t002:** Lipid and fatty acids composition of *Crepis vesicaria* subsp. *taraxacifolia* leaves (mean ± SD; *n* = 3).

Composition	Raw Matter	Dry Matter
Fatty acids, total polyunsaturated (mg/100 g)	402.840 ± 0.146	2786.53 ± 1.01
Fatty acids, total saturated (mg/100 g)	159.820 ± 0.207	1105.48 ± 1.44
Fatty acids, total monounsaturated (mg/100 g)	123.710 ± 0.063	855.75 ± 0.44
α-Linolenic acid (C18:3n-3) (mg/100 g)	343.240 ± 0.065	2374.30 ± 0.57
Linoleic acid (C18:2n-6) (mg/100 g)	59.600 ± 0.084	412.23 ± 0.45
Oleic acid (C18:1n-9) (mg/100 g)	60.490 ± 0.087	418.43 ± 0.58
Palmitic acid (C16:0) (mg/100 g)	108.750 ± 0.004	752.26 ± 0.56
Gondoic acid (C20:1) (mg/100 g)	63.220 ± 0.086	437.32 ± 0.59
Arachidic acid (C20:0) (mg/100 g)	17.490 ± 0.091	121.00 ± 0.63
Stearic acid (C18:0) (mg/100 g)	21.520 ± 0.047	148.84 ± 0.60
Margaric acid (C17:0) (mg/100 g)	12.050 ± 0.081	83.38 ± 0.33
Lipids (g/100 g)	0.690 ± 0.004	4.78 ± 0.03

**Table 3 ijerph-18-00151-t003:** Minerals and heavy metal composition of *Crepis vesicaria* subsp. *taraxacifolia* leaves (mean ± SD; *n* = 3).

Composition		Raw Matter	Dry Matter
Ash (g/100 g)		2.740 ± 0.007	18.94 ± 0.05
Minerals	Potassium (mg/100 g)	591.290 ± 0.058	4090.07 ± 0.31
	Calcium (mg/100 g)	309.930 ± 0.090	2143.84 ± 0.62
	Sodium (mg/100 g)	76.780 ± 0.084	531.12 ± 0.42
	Phosphorus (mg/100 g)	59.910 ± 0.074	412.33 ± 0.51
	Magnesium (mg/100 g)	45.460 ± 0.066	314.46 ± 0.34
	Zinc (mg/100 g)	0.860 ± 0.012	5.97 ± 0.09
	Manganese (mg/100 g)	0.830 ± 0.006	5.71 ± 0.04
	Iron (mg/100 g)	0.590 ± 0.003	4.05 ± 0.02
	Copper (mg/100 g)	0.420 ± 0.012	2.89 ± 0.09
	Nickel (mg/100 g)	0.110 ± 0.011	0.79 ± 0.08
	Chromium (mg/100 g)	0.020 ± 0.003	0.11 ± 0.02
Heavy metals	Cadmium (µg/100 g)	19.300 ± 0.076	133.52 ± 0.53
	Lead (µg/100 g)	2.300 ± 0.094	15.92 ± 0.65
	Mercury (µg/100 g)	0.050 ± 0.002	0.36 ± 0.01

**Table 4 ijerph-18-00151-t004:** Antioxidant activity of ethanolic extract (Cv 70% EtOH) from *Crepis vesicaria* L. subsp. *taraxacifolia*.

	IC_50_ (µg/mL)	TE *
DPPH^●^	26.20 ± 1.86	111.980 ± 0.041
ABTS^●^ (pH = 7)	18.92 ± 2.24	21.670 ± 0.012
FRAP	-	0.678 ± 0.168

* TE (Trolox Equivalent): Amount of the samples (µg/mL) that has the same anti-radical activity of Trolox 1 mM. The results are expressed as mean ± SD of three independent experiments.

**Table 5 ijerph-18-00151-t005:** Compounds identified in Cv 70% ethanol extract by HPLC-PDA-ESI/MS ^n^.

Compound	Partial Identification	R_t_ (min.)	λ_max._ by HPLC/PDA (nm)	[M-H]^−^	MS ^2^	MS ^3^
**1**	Caffeic acid [44]	13.54	238, 251, 291 sh, 299 sh, 328	179	[179]: 135 (100)	[179 135]: 151 (13), 135 (61), 125 (11), 107 (24), 91 (100)
**2**	Quinic acid [44]	20.33	238, 253, 291 sh, 300 sh, 326	191	[191]: 173 (82), 171 (24), 147 (12), 127 (100), 111 (45), 109 (27), 93 (46), 87 (14), 85 (66)	---
**3**	Chicoric acid isomer [45]	25.59	238, 251, 291 sh, 299 sh, 329	473	[473]: 311 (100), 293 (80)	[473 311]: 179 (62), 149 (100)
**4**	Chicoric acid isomer [45]	26.12	238, 251, 292 sh, 300 sh, 329	473	[473]: 311 (100), 293 (80)	[473 311]: 179 (58), 149 (100)
**5**	Chicoric acid isomer [45]	28.47	238, 251, 292 sh, 299 sh, 330	473	[473]: 311 (100), 293 (80)	[473 311]: 179 (61), 149 (100)
**6**	Feruloyl hexosylpentoside [46]	32.77	238, 253, 292 sh, 299 sh, 329	487	[487]: 325 (100), 307 (46), 293 (77)	[487 325]: 193 (100)

Identification based on the UV-Vis spectra, molecular weight and fragmentation patterns, which are according to authors referred.

## Data Availability

All data is available based on “MDPI Research Data Policies” at https://www.mdpi.com/ethics.

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
