# Peer review of "Crepis vesicaria* L. subsp. *taraxacifolia* Leaves: Nutritional Profile, Phenolic Composition and Biological Properties"

_ijerph, 2020, doi:10.3390/ijerph18010151_

Round 1
Reviewer 1 Report
The manuscript entitled "Crepis vesicaria L. subsp. Taraxacifolia leaves: nutritional profile, phenolic composition and biological properties" addresses a phytochemical and nutritional investigation of a species that is used as food and as a medicinal plant. In addition, it evaluates the antioxidant and anti-inflammatory capacity of extracts of C. vesicaria.
The article brings relevant information about the nutritional and chemical potential of the species as well as the biological activities evaluated, being characterized as a pioneering study.
The manuscript, under my assessment, can be accepted for publication after reviewing and clarifying the questions below.
Line 30. delete dot after assays.
Line 66. the popular name of this species under study must be included. It remained to report the ethnomedicinal information, since in the abstract this information is confirmed.
Line 77. Provide registration number or code for the exsiccatae deposit.
Line 77. The leaves were dried how? In the sun? the shadow ? in a circulating air kiln?
Line 79. Authors should further specify how they concentrated the extract obtained from the infusion. What temperature did you use in the rotavapor bath?
Line 103. Detail methodology by HPLC and detector used to quantify sugars.
Line 151. How was the obtained IC50 of DDPH?
Line 216. Put the species name in italics.
Line 268. Correct the name of the plant species
Line 298. How absorption of which concentration of the samples tested by both DPPH and ABTS in the TEAC calibration curve was interpolated?
Line 306. Improve the chromatogram figure.
Line 315. Improve the arrangement of the information in Table 3 and explain which ion from MS2 was used to make MS3.
Line 315. Improve the arrangement of the information in Table 3. and explain which ion from MS2 was used to make MS3.
Line 333. The authors report on how the isomers of chicoric acid elute in reverse phases by HPLC, however it is necessary, even when quoting, to be clear in what chromatographic conditions this form of elution occurs. If in the reference cited are the same solvents used in your experiments, type of column and pH of the solvents used. Otherwise, this affirmation not present in the text should not be placed.
Line 344. If the authors use a commercial standard of chicorc acid to perform the quantification by HPLC-DAD in the extract Cv 70%, why not use this standard to affirm at least one of the 3 isomers present in the extracts identified by HPLC-MS? And which of the 3 isomers is this commercial standard? This information must appear in the text.
Author Response
Revised manuscript submission ijerph-1020131
Dear Reviewer #1,
First of all, authors would like to thank you very much for the comments. Based on your comments, the manuscript was improved according to the suggestions.
Thus, and on behalf of all co-authors, I send you a revised manuscript, in addition to the authors’ response to your comments.
Authors have considered the comments presented by all the Reviewers and detailed information was provided in order to respond to the Reviewers comments (please see the response to your comments below and the attached “ijerph-1020131 - Revised manuscript” file where “track changes” function was used to highlight revisions).
Authors hope that the revised manuscript can be accepted for publication in the International Journal of Environmental Research and Public Health.
Response to Reviewer 1
The manuscript entitled "Crepis vesicaria L. subsp. Taraxacifolia leaves: nutritional profile, phenolic composition and biological properties" addresses a phytochemical and nutritional investigation of a species that is used as food and as a medicinal plant. In addition, it evaluates the antioxidant and anti-inflammatory capacity of extracts of C. vesicaria.
The article brings relevant information about the nutritional and chemical potential of the species as well as the biological activities evaluated, being characterized as a pioneering study.
The manuscript, under my assessment, can be accepted for publication after reviewing and clarifying the questions below.
Line 30. delete dot after assays.
R: correction was done in the manuscript
Line 66. the popular name of this species under study must be included. It remained to report the ethnomedicinal information, since in the abstract this information is confirmed.
R: The information requested was added in the manuscript
Line 77. Provide registration number or code for the exsiccatae deposit.
R: The registration number was added to the manuscript.
Line 77. The leaves were dried how? In the sun? the shadow? in a circulating air kiln?
R: The drying process was included in the manuscript
Line 79. Authors should further specify how they concentrated the extract obtained from the infusion. What temperature did you use in the rotavapor bath?
R: The temperature was added to the manuscript.
Line 103. Detail methodology by HPLC and detector used to quantify sugars
.
R: HPLC methodology details, including detector, were included in the manuscript
Line 151. How was the obtained IC50 of DDPH?
R: the requested information was added to the manuscript.
Line 216. Put the species name in italics.
R: the species name in italic was put in the manuscript
Line 268. Correct the name of the plant species
R: the name of the plant species was corrected in the manuscript
Line 298. How absorption of which concentration of the samples tested by both DPPH and ABTS in the TEAC calibration curve was interpolated?
R: It was answered in the manuscript.
Line 306. Improve the chromatogram figure.
R: The authors improved the resolution of the chromatogram in the manuscript.
Line 315. Improve the arrangement of the information in Table 3 and explain which ion from MS2 was used to make MS3.
R: The table was rearranged and the parent m/z peaks were identified (see table 5 of the revised manuscript) .
Line 333. The authors report on how the isomers of chicoric acid elute in reverse phases by HPLC, however it is necessary, even when quoting, to be clear in what chromatographic conditions this form of elution occurs. If in the reference cited are the same solvents used in your experiments, type of column and pH of the solvents used. Otherwise, this affirmation not present in the text should not be placed.
R: The authors thank the revisor for the suggestion and decided to withdraw the affirmation.
Line 344. If the authors use a commercial standard of chicoric acid to perform the quantification by HPLC-DAD in the extract Cv 70%, why not use this standard to affirm at least one of the 3 isomers present in the extracts identified by HPLC-MS? And which of the 3 isomers is this commercial standard? This information must appear in the text.
R: The information requested was added to the manuscript.

Reviewer 2 Report
Dear authors:
The manuscript “Crepis vesicaria L. subsp. taraxacifolia leaves: nutritional profile, phenolic composition and biological properties” reports the characterization of a promising raw material that could be potentially used as a functional ingredient. The manuscript brings interesting data and a decent discussion of the results. However, some points of improvement should be considered in a future re-submission, as outlined below. Therefore, I recommend minor revisions.
General:
English grammar needs to be revised throughout the whole manuscript.
Abstract:
Line 25: Replace “posteriorly” with “subsequently.”
Lines 27-28: Include the total amount of carbohydrates, minerals, protein, and lipids.
Line 29: Include a couple more phenolics that were detected.
Line 33: Replace “food functional” with “functional food.”
Introduction:
Line 51: antiproliferative effect against what type of cell? Antioxidant activity measured by which methods?
Line 54: Which type of tannins?
Lines 62-65: There are a number of ways phenolics can work to reduce oxidative stress. Please, expand this topic by discussing those several mechanisms.
Material and Methods:
Line 77: Replace “posteriorly” with “subsequently.”
Item 2.1: How were the extraction conditions determined?
Item 2.1: Please, explicitly state that at the end of the extraction process, the extracts were rich in soluble phenolics.
Item 2.3.1: How were the results expressed? Did the authors use a calibration curve?
Line 186: What were the concentrations tested?
Item 2.4.1: Any reference for this method?
Item 2.4.2: Reference?
Results and discussion:
Comment: Table 1should be reorganized. The compounds should be sub-grouped (e.g., minerals, fatty acids, etc) and the components should be organized in a descending order according to their amounts.
Lines 217-218: Any organic material is prone to deterioration. Please, be more specific.
Lines 232-234: Judging by this result, don’t the authors think that it would have been wise to also measure insoluble-bound phenolics compounds? Please, make a comment on the subject in the paper. In fact, the authors should at least acknowledge the existence of bioactive phenolic compounds. For that they could mention the following reference.
Albishi, T., Banoub, J. H., de Camargo, A. C., & Shahidi, F. (2019). Wood extracts as unique sources of soluble and insoluble-bound phenolics: reducing power, metal chelation and inhibition of oxidation of human LDL-cholesterol and DNA strand scission. Journal of Food Bioactives, 8. https://doi.org/10.31665/JFB.2019.8211
Line 261: Based on those findings, would the authors recommend the consumption of this plant?
Lines 286-288: This should be in the “Material and methods” section.
Figure 1: Peaks 3, 4, and 5 are faint.
Question: Why only chicoric acid was quantified?
Table 2 and Line 300, the authors should use “TE, Trolox Equivalent” which is the most appropriated form.
Considering that the authors do not report the IC50 for FRAP, do they think their extract present reducing power?
The authors did not use any cell-based assay (please see de Camargo et al., https://doi.org/10.1016/j.foodchem.2019.03.145) but the literature describes that phenolic extracts bearing higher antiradical activity towards DPPH and ABTS also show higher inhibition of NF‐κB activation which is mediated by oxidative species (please see Falcão et al. https://doi.org/10.1111/jfbc.13018). Therefore, the authors could justify the use of DPPH and ABTS as well as the selection of 70 % EtOH extract by mentioning the study by Falcão et al. https://doi.org/10.1111/jfbc.13018.
Lines 320-342 – The authors did not mention that they have confirmed the identities of the molecules. Therefore they should always state that …. compound XX was TENTATIVELY identified as XX.
It is important to mention that all references mentioned here were purely based on COPE Ethical Guidelines for Peer Reviewers (http://publicationethics.org/files/Ethical_guidelines_for_peer_reviewers_0.pdf) that states that “suggestions must be based on valid academic or technological reasons.”
For example, Dr. Fereidoon Shahidi, the senior author of some manuscripts suggested here, is we well known in the field of antioxidants. First in citations according to google scholar (https://scholar.google.com/citations?hl=en&view_op=search_authors&mauthors=label:antioxidants). Therefore, suggesting his studies is self-explanatory.
Author Response
Revised manuscript submission ijerph-1020131
Dear Reviewer #2,
First of all, authors would like to thank you very much for the comments. Based on your comments, the manuscript was improved according to the suggestions.
Thus, and on behalf of all co-authors, I send you a revised manuscript, in addition to the authors’ response to your comments.
Authors have considered the comments presented by all the Reviewers and detailed information was provided in order to respond to the Reviewers comments (please see the response to your comments below in blue and the attached “ijerph-1020131 - Revised manuscript” file where “track changes” function was used to highlight revisions).
Authors hope that the revised manuscript can be accepted for publication in the International Journal of Environmental Research and Public Health.
Response to Reviewer 2
The manuscript “Crepis vesicaria L. subsp. taraxacifolia leaves: nutritional profile, phenolic composition and biological properties” reports the characterization of a promising raw material that could be potentially used as a functional ingredient. The manuscript brings interesting data and a decent discussion of the results. However, some points of improvement should be considered in a future re-submission, as outlined below. Therefore, I recommend minor revisions.
General:
English grammar needs to be revised throughout the whole manuscript.
R: The English grammar was revised
Abstract:
Line 25: Replace “posteriorly” with “subsequently.”
R: Corrected in the manuscript
Lines 27-28: Include the total amount of carbohydrates, minerals, protein, and lipids.
R: The information requested was included in the manuscript
Line 29: Include a couple more phenolics that were detected.
R: More phenolic compounds were included in the abstract.
Line 33: Replace “food functional” with “functional food.
”
R: Was replaced in the abstract.
Introduction:
Line 51: antiproliferative effect against what type of cell? Antioxidant activity measured by which methods?
R: The information was included in the introduction.
Line 54: Which type of tannins?
R: The type of tannins was included in the introduction.
Lines 62-65: There are a number of ways phenolics can work to reduce oxidative stress. Please, expand this topic by discussing those several mechanisms.
R: This topic was developed in the introduction
Material and Methods:
Line 77: Replace “posteriorly” with “subsequently.”
R: Replaced in the manuscript
Item 2.1: How were the extraction conditions determined?
R: The extraction conditions optimized by preparing extracts with different solvent concentrations and evaluating the antioxidant activity. The infusion was prepared according the medicinal uses. These informations were added in the manuscript.
Item 2.1: Please, explicitly state that at the end of the extraction process, the extracts were rich in soluble phenolics.
R: The requested information was added to the manuscript.
Item 2.3.1: How were the results expressed? Did the authors use a calibration curve?
R: The requested information was added to the manuscript.
Line 186: What were the concentrations tested?
R: The concentrations tested were added to the manuscript.
Item 2.4.1: Any reference for this method?
R: The reference was added to the manuscript.
Item 2.4.2: Reference?
R: The reference was added to the manuscript.
Results and discussion:
Comment: Table 1 should be reorganized. The compounds should be sub-grouped (e.g., minerals, fatty acids, etc) and the components should be organized in a descending order according to their amounts.
R: The information was reorganized according to your suggestion.
Lines 217-218: Any organic material is prone to deterioration. Please, be more specific.
R: We thank the reviewer for highlighting this point, which is indeed relevant. We have added new information to clarify this aspect (please see lines 217-221, p6).
Lines 232-234: Judging by this result, don’t the authors think that it would have been wise to also measure insoluble-bound phenolics compounds? Please, make a comment on the subject in the paper. In fact, the authors should at least acknowledge the existence of bioactive phenolic compounds. For that they could mention the following reference.
Albishi, T., Banoub, J. H., de Camargo, A. C., & Shahidi, F. (2019). Wood extracts as unique sources of soluble and insoluble-bound phenolics: reducing power, metal chelation and inhibition of oxidation of human LDL-cholesterol and DNA strand scission. Journal of Food Bioactives, 8. https://doi.org/10.31665/JFB.2019.8211
R: We thank for the suggestion. That comment was included in the manuscript, as well as the reference.
Line 261: Based on those findings, would the authors recommend the consumption of this plant?
R: We consider that Crepis vesicaria subsp. taraxacifolia is a resource of great interest in nutritional terms. However, in view of the risk of heavy metal contamination, which is also a transversal problem to most foods, it is essential that when used for food purposes the specimens are harvested in unpolluted areas. That is, collected from soils free from heavy metal contamination. In particular, regarding cadmium, the heavy metal that was detected in greater quantities, the use of fertilizers produced from phosphate ores, as well as the inappropriate disposal of waste containing cadmium must be avoided as much as possible to mitigate this problem.
Lines 286-288: This should be in the “Material and methods” section.
R: This part was relocated in Material and methods section.
Figure 1: Peaks 3, 4, and 5 are faint.
R: The authors tryed to improve the chromatogram quality.
Question: Why only chicoric acid was quantified?
R: We only quantified the chicoric acid because this was the main constituent in the sample and, according to the literature, many of the biological activities evaluated could be attributed to this compound. (We added this comment in the section 3.4.)
Table 2 and Line 300, the authors should use “TE, Trolox Equivalent” which is the most appropriated form.
R: We corrected this designation in the manuscript.
Considering that the authors do not report the IC50 for FRAP, do they think their extract present reducing power?
R: It was answered in the manuscript. The extract present reducing power because the TE value in ug/mL of the extract is lower than the corresponding value of concentration of trolox 1 mM.
The authors did not use any cell-based assay (please see de Camargo et al., https://doi.org/10.1016/j.foodchem.2019.03.145) but the literature describes that phenolic extracts bearing higher antiradical activity towards DPPH and ABTS also show higher inhibition of NF‐κB activation which is mediated by oxidative species (please see Falcão et al. https://doi.org/10.1111/jfbc.13018). Therefore, the authors could justify the use of DPPH and ABTS as well as the selection of 70 % EtOH extract by mentioning the study by Falcão et al. https://doi.org/10.1111/jfbc.13018.
R: The authors thank to the revisor the suggestion and included an explanation on this subject.
Lines 320-342 – The authors did not mention that they have confirmed the identities of the molecules. Therefore they should always state that …. compound XX was TENTATIVELY identified as XX.
R: Correct in the manuscript.
It is important to mention that all references mentioned here were purely based on COPE Ethical Guidelines for Peer Reviewers (http://publicationethics.org/files/Ethical_guidelines_for_peer_reviewers_0.pdf) that states that “suggestions must be based on valid academic or technological reasons.”
For example, Dr. Fereidoon Shahidi, the senior author of some manuscripts suggested here, is we well known in the field of antioxidants. First in citations according to google scholar (https://scholar.google.com/citations?hl=en&view_op=search_authors&mauthors=label:antioxidants). Therefore, suggesting his studies is self-explanatory.

Reviewer 3 Report
The authors have presented an innovative findings in this manuscript, however, there are few typos and grammatical errors that will need to be corrected.
Also, Table should be section into the different nutritional and phenolic compounds determined.
For example. Table 1. Lipid and fatty acid composition of Crepis vesicaria L. subsp. taraxacifolia leave
Table 2. Phenolic compounds composition of Crepis vesicaria L. subsp. taraxacifolia leave
Table 3. Essential and heavy metal composition of Crepis vesicaria L. subsp. taraxacifolia leaves
Table 4. Nutritive (carbohydrates, proteins) content of Crepis vesicaria L. subsp. taraxacifolia leaves
This way authors can insert these tables under each heading and present the results. The current presentation of the results is highly confusing.
Author Response
Revised manuscript submission ijerph-1020131
Dear Reviewer #3,
First of all, authors would like to thank you very much for the comments. Based on your comments, the manuscript was improved according to the suggestions.
Thus, and on behalf of all co-authors, I send you a revised manuscript, in addition to the authors’ response to your comments.
Authors have considered the comments presented by all the Reviewers and detailed information was provided in order to respond to the Reviewers comments (please see the response to your comments below in blue and the attached “ijerph-1020131 - Revised manuscript” file where “track changes” function was used to highlight revisions).
Authors hope that the revised manuscript can be accepted for publication in the International Journal of Environmental Research and Public Health.
Response to Reviewer 3
The authors have presented an innovative findings in this manuscript, however, there are few typos and grammatical errors that will need to be corrected.
Also, Table should be section into the different nutritional and phenolic compounds determined.
For example. Table 1. Lipid and fatty acid composition of Crepis vesicaria L. subsp. taraxacifolia leave
Table 2. Phenolic compounds composition of Crepis vesicaria L. subsp. taraxacifolia leave
Table 3. Essential and heavy metal composition of Crepis vesicaria L. subsp. taraxacifolia leaves
Table 4. Nutritive (carbohydrates, proteins) content of Crepis vesicaria L. subsp. taraxacifolia leaves
This way authors can insert these tables under each heading and present the results. The current presentation of the results is highly confusing
.
R: The results were divided into several tables according to your suggestion.

Reviewer 4 Report
The manuscript "Crepis vesicaria L. subsp. taraxacifolia leaves: nutritional profile, phenolic composition and biological properties" comprehensively characterize the chemical composition of an aqueous and an ethanolic extract from the leaves of Crepis vesicaria L. and the potential antioxidant and anti-inflammatory properties of the ethanolic extract.
The authors did a tremendous analytical job, but the manuscript did not show them correctly. The authors should rewrite some parts to present their results to the scientific community better.
As it is, the manuscript is not ready to be published.
The reviewer comments are listed below:
- In the abstract, no sections are needed. Eliminate them.
- Also, in the abstract, rewrite HPLC.
- Table 1 should be divided into several tables to ease the shear of values.
- Also, in Table 1. the authors should carefully revise the use of significant figures.
- Section 3.1. could be divided into subsections to discuss the results better.
- The quality of the chromatogram is really low.
- Why were only the properties of the ethanolic extract measured?
- The authors should discuss cell viability first and anti-inflammatory effects.
- The conclusion section should avoid giving or summarising the results. Rewrite it.
- The authors should rewrite the funding and acknowledgments sections.
- Eliminate highlights.
Author Response
Revised manuscript submission ijerph-1020131
Dear Reviewer #4,
First of all, authors would like to thank you very much for the comments. Based on your comments, the manuscript was improved according to the suggestions.
Thus, and on behalf of all co-authors, I send you a revised manuscript, in addition to the authors’ response to your comments.
Authors have considered the comments presented by all the Reviewers and detailed information was provided in order to respond to the Reviewers comments (please see the response to your comments below in blue and the attached “ijerph-1020131 - Revised manuscript” file where “track changes” function was used to highlight revisions).
Authors hope that the revised manuscript can be accepted for publication in the International Journal of Environmental Research and Public Health.
Response to Reviewer 4
The manuscript "Crepis vesicaria L. subsp. taraxacifolia leaves: nutritional profile, phenolic composition and biological properties" comprehensively characterize the chemical composition of an aqueous and an ethanolic extract from the leaves of Crepis vesicaria L. and the potential antioxidant and anti-inflammatory properties of the ethanolic extract.
The authors did a tremendous analytical job, but the manuscript did not show them correctly. The authors should rewrite some parts to present their results to the scientific community better.
As it is, the manuscript is not ready to be published.
The reviewer comments are listed below:
- In the abstract, no sections are needed. Eliminate them.
R: the sections were eliminated
- Also, in the abstract, rewrite HPLC.
R: the error was corrected.
- Table 1 should be divided into several tables to ease the shear of values.
R: The table was divided according to suggestion.
- Also, in Table 1. the authors should carefully revise the use of significant figures.
R: The values were corrected for the appropriate significant figures.
- Section 3.1. could be divided into subsections to discuss the results better.
R: The section 3.1. was divided according to your suggestion
- The quality of the chromatogram is really low
R: The authors tried to improve the quality of the chromatogram.
- Why were only the properties of the ethanolic extract measured?
R: The answer to this question was included in section 3.2 (lines 434 and 443)
- The authors should discuss cell viability first and anti-inflammatory effects.
R: The discussion was reordered.
- The conclusion section should avoid giving or summarising the results. Rewrite it
R: The conclusion was rewritten
- The authors should rewrite the funding and acknowledgments sections.
R: The funding and acknowledgments were rewritten.
- Eliminate highlights.
R: The highlights were eliminated.

Round 2
Reviewer 1 Report
The manuscript entitled "Crepis vesicaria L. subsp. Taraxacifolia leaves: nutritional profile, phenolic composition and biological properties" was significantly improved and the authors answered all the questions asked. Thus, the manuscript can be accepted for publication.
Reviewer 4 Report
The manuscript "Crepis vesicaria L. subsp. taraxacifolia leaves: nutritional profile, phenolic composition and biological properties" comprehensively characterize the chemical composition of an aqueous and an ethanolic extract from the leaves of Crepis vesicaria L. and the potential antioxidant and anti-inflammatory properties of the ethanolic extract.
After the manuscript revision, I consider the authors responded to all reviewers' comments and changed their mansucript according to them.
Then, as it is, the manuscript is now ready to be published.